# A Fine-Grained User-Divided Privacy-Preserving Access Control Protocol in Smart Watch

**DOI:** 10.3390/s19092109

**Published:** 2019-05-07

**Authors:** Liming Fang, Minghui Li, Lu Zhou, Hanyi Zhang, Chunpeng Ge

**Affiliations:** 1College of Computer Science and Technology, Nanjing University of Aeronautics and Astronautics, No. 29 Yudao Street, Nanjing 210016, China; fangliming@nuaa.edu.cn (L.F.); leemh@nuaa.edu.cn (M.L.); kyouichi@nuaa.edu.cn (H.Z.); gecp@nuaa.edu.cn (C.G.); 2Key Laboratory of Computer Network Technology of Jiangsu Province, Nanjing 210096, China; 3Division of Computer Science, University of Aizu, Aizuwakamatsu 965-8580, Japan

**Keywords:** smart watch, privacy preservation, re-encryption, homomorphic computation, attribute-based encryption

## Abstract

A smart watch is a kind of emerging wearable device in the Internet of Things. The security and privacy problems are the main obstacles that hinder the wide deployment of smart watches. Existing security mechanisms do not achieve a balance between the privacy-preserving and data access control. In this paper, we propose a fine-grained privacy-preserving access control architecture for smart watches (FPAS). In FPAS, we leverage the identity-based authentication scheme to protect the devices from malicious connection and policy-based access control for data privacy preservation. The core policy of FPAS is two-fold: (1) utilizing a homomorphic and re-encrypted scheme to ensure that the ciphertext information can be correctly calculated; (2) dividing the data requester by different attributes to avoid unauthorized access. We present a concrete scheme based on the above prototype and analyze the security of the FPAS. The performance and evaluation demonstrate that the FPAS scheme is efficient, practical, and extensible.

## 1. Introduction

With the advent of mini smart independent chips and the popularity of the mobile Internet of Things (IoT), smart wearable devices are gradually becoming popular in people’s lives [1]. A smart watch can realize functions including calling, social entertainment, human health monitoring, etc [2]. It i shown that smart wearable devices are fashionable in the smart devices market in Figure 1 from market research [3]. However, there are some potential risks that an attack can make a smart wearable device become a monitoring device to collect private information of the wearer illegally due to the irregular design flaws [4]. In 2017, the Norwegian Consumer Council pointed out that some children’s watches have vulnerabilities, such as transmitting and storing data without encryption, in the report [5]. This means that strangers can use basic hacking techniques to track children’s actions or falsify the actual location. Hence, the secure performance of smart watches should be improved.

The working mechanism of the whole smart watch system is divided into three parts: Pairing between the smart watch and the App, uploading data from the smart watch, and accessing control of the private data. Figure 2 illustrates the information interaction process of the smart wearable devices and corresponding management tools. First, a smart watch pairs with the smartphone and initializes the parameters. After the smart watch connects to the network, some information, such as the calls, photos, and music on the mobile phone, can be synchronized to the smart watch. Next, the personal health information collected by the smart watch is uploaded to the cloud. Then, the cloud sends the computed data to the smartphone and another management App, which imposes a restriction on accessing data according to different users’ authorities.

For the information interaction mechanism in Figure 2, there are some vulnerabilities that can be exploited, including the connection process, data processing in the cloud, and data accessing by a third party user. To be specific, a malicious connection happens between an illegal device and an illegal App due to the lack of authentication. There are researchers who claim that a critical cryptographic vulnerability has been found affecting Bluetooth that could allow an unauthenticated user to access the data [6]. Moreover, if the user uploads plaintext data, the cloud may directly obtain the corresponding information. Even if ciphertext is sent to cloud, the cloud server may adopt its powerful computing power to decrypt the private ciphertext. Furthermore, the cloud server does not have a perfect identification system, so a user A also can operate other unbound smart watches through his/her App, which leads to an unauthorized loophole [7]. An attacker utilizes the weak point to speculate about users’ behavior, and the leads to leakage of private data.

In order to solve the above problem, a security scheme is proposed. Because of the openness of wireless communication in smart devices, it is hard to ensure security and protect privacy [8]. The existing research is mainly aimed at increasing the smart watch extension function [9] and preventing the motion sensor from leaking user activity status [10]. There are no other authentication measures to confirm except the personal identification number (PIN) in the Bluetooth protocol [11]. Moreover, the existing cloud-untrusted solution in the Internet of Things does not take into account the fine-grained access control [12,13,14]. In addition, private data are easily acquired by the cloud [15,16]. Considering the lightweight nature of IoT devices, some smart watches are low computational devices [17]. Therefore, the over-complex encryption schemes [18,19] are not applicable to the smart wearable devices. Therefore, we need to solve the challenge for a security scheme in smart wearable devices:How do we resist the illegal device connection, in case the Bluetooth hacking vulnerability affects identify authentication?How do we resist the information leakage in data processing by an non-credible cloud?How do we realize secure information sharing with different people reducing the authentication steps as much possible?Is it suitable for low computational devices?

**Motivation.** To overcome the above challenges, we propose the solutions as follows:Each device has the unique codes media access control (MAC) and universally-unique identifier (UUID). We can utilize and save the unique codes to avoid the illegal connection of malicious devices.It is necessary to calculate and convert the encrypted data at the untrusted third party. Therefore, homomorphic computation [20] and re-encryption [18] are used to secure multi-party computation and ciphertext conversion.In order to implement information sharing, it is a good idea to use cipher-policy attribute-based encryption (CP-ABE) in the public cloud [21]. The fine-grained access control structure can be decided by the encryption part [22]. Moreover, this fine-grained access control not only refers to restrictions on the person accessing private data, but also to partial sharing of personal information.We also consider the differences in computing power of the smart watch. To be suitable for a low-computing power device, we make the cloud undertake the complex encryption and homomorphic computation. The smart watch is only responsible for some basic computations.

Based on the above observations, we propose a privacy protection scheme in a smart watch called FPAS. The core idea of FPAS is using homomorphic computation, re-encryption, and attribute encryption to ensure the security in an untrusted cloud environment. The security goals include fine-grained personal information sharing without leakage.

**Contributions.** Most existing security solutions are based on the assumption that the cloud is completely trusted. The current schemes have no balance between privacy preserving and information sharing; at the same time, they often ignore the low-computing power devices. Hence, we propose a fine-grained user-divided with privacy protection scheme in a smart watch to overcome the above challenges in this paper. It supports encrypted data comparing, secure fine-grained access control, and ciphertext conversion in the smart watch system. Specifically, the contributions can be summarized as follows.

We propose a framework for protecting personal private information in a smart watch, even though the cloud server is untrusted. The framework can compute and share data confidentially without leaking private user information.FPAS delivers a part of the complex operations to the cloud without leaking private data, which reduces the device’s computing resource consumption.Our solution reduces the risk of illegal application connection between the smart watch and the App.Our proposed solution has good expansibility such that it can ignore the different terminals.

**Organization.** The rest of paper is organized as follows. In Section 2, we introduce the related work. Section 3 gives a brief overview of the methodology. Section 4 shows our scheme. The proposed scheme is proven to be secure in Section 5. We give a performance analysis in Section 6. Conclusions are finally drawn in Section 7.

## 2. Related Work

Some works have proven that there are potential security issues in smart wearable devices. Lee et al. [23] discovered that copying private user data during the pairing of watches and phones leads to the disclosure of private information. Do et al. [24] found a way to obtain sensitive data from Android-based smart watches and to describe the type and location of sensitive data on smart watches.

For solving the problems of personal privacy disclosure, researchers have focused on different encryption algorithms, such as homomorphic encryption, re-encryption, and public key encryption with keyword search technology. Khedr et al. [25] proposed a scheme for sensitive medical data sharing using homomorphic encryption for computing encrypted data without decryption. Fang et al. [26] improved the public key encryption with a keyword search scheme to implement a secure ciphertext search on a non-trusted server to prevent information leakage and proved the IND-SCF-CKCAsecurity of the solution. Guo et al. [27] used logistic regression to design a medical data prediction system in a cloud environment without revealing user privacy. The scheme used homomorphic encryption technology to calculate private ciphertext. Rohloff et al. [19] proposed an end-to-end protection framework for medical devices that uses homomorphic encryption and proxy re-encryption to provide private processing for data collected by devices. Petrlic et al. [28] proposed a multi-party authorization scheme that does not require trusted third parties to protect privacy. The scheme was based on attribute encryption and anonymous covert outsourcing calculations. Zhang et al. [13] combined the matrix generation communicate key and the physical layer key to implement secure node interaction in smart devices. Li et al. [29] designed a keyword-based search scheme based on attribute encryption, and the cloud service provider partially decrypted the data to prevent user information from leaking. Ding et al. [30] used Paillier homomorphic encryption and attribute-based encryption to implement private data protection in a cloud environment with multiple operations and flexible access control. Liu et al. [31] proposed a fine-grained e-healthcare record access control scheme with encoding in linear secret sharing.

In order to ensure secure information sharing, a large amount of work on access control and authorization has been proposed. Yan et al. [32] proposed a scheme for data access control in cloud computing based on attribute encryption and proxy re-encryption by trusted organizations. Ge et al. [33] proposed a key-policy attribute-based proxy re-encryption (KP-ABPRE), which uses a proxy re-encryption scheme with fine-grained access control with attributes to effectively protect private data in the cloud. Ge et al. [34] also proposed the identity-based proxy re-encryption with chosen ciphertext security, in which each re-encryption key is associated with an access tree. This solution guarantees the privacy of information sharing in cloud computing. Fang et al. [35] proposed fuzzy conditional broadcast proxy re-encryption (FC-BPRE) for shared server and user data security in the cloud. Only when a set of conditions is satisfied can ciphertext conversion be performed. Li et al. [36] introduced a novel identity-based encryption to implement a secret sharing scheme that supports decrypting two ciphertexts by different keys. This work has solved the problem of third party access rights’ control in the cloud server. However, in low-computing power devices such as smart watches, complex calculations are often not efficient. Therefore, they cannot be applied to the scene of smart watch data sharing.

Based on the above deficiencies, we propose a privacy protection scheme for fine-grained access control of information content and access rights when the cloud is not trusted. The solution is compatible with devices that have different computing power.

## 3. Preliminaries

We introduce the background technology in this section. Table 1 summarizes the symbols used in this paper.

### 3.1. Ciphertext-Policy Attribute-Based Encryption

In ciphertext-policy attribute-based encryption [37,38], the user’s key is specified by a set of attributes, and the encrypted data are associated with a policy. Only when decrypting the party’s attributes specifying the access tree can he/she decrypt the ciphertext. The steps of the algorithm are as follows:

CP–ABE–Setup(λ) : This algorithm chooses a security parameter λ and generates a public key PK and a master key MK.

CP–ABE–Encrypt(PK,m,A): The algorithm encrypts a message *m* under the tree access structure *A* and outputs the ciphertext CT.

CP–ABE–KeyGen(MK,S): The key generation algorithm inputs the master key MK and an attribute set *S*, then outputs a private key SK that is associated with the attribute set.

CP–ABE–Decrypt(PK,CT,SK): The decryption algorithm takes as input the public parameters PK, a ciphertext CT that includes an access control tree *A*, and a private key SK, which is for a set *S* of attributes. Only if satisfying the access structure, the algorithm decrypts the ciphertext and returns a message *m*.

The user implements the fine-grained division of access rights by setting the access structure. The algorithms are widely used in access control of cloud services.

### 3.2. Proxy Re-Encryption

Proxy re-encryption allows a not fully-dependable third party to change encrypted data. This means that data encrypted by Alice’s public key are converted to new encrypted data that may be decrypted by Ben’s private key. In this scheme, the proxy, namely the cloud computing server, cannot acquire the underlying message. Moreover, a proxy re-encryption scheme enables transforming one’s ciphertext to another’s ciphertext to achieve information sharing without leaking user’s secret key [33,39,40].

## 4. Definition of Our Scheme

### 4.1. Attack Model

Before introducing our scheme, we summarize several possible attacks on smart watches. Attacker A can implement the following attack scenarios:Illegal application connection: Roman Unuchek [41] found that many common smart wearable devices allow the third party App to invisibly connect. This process can execute commands and even obtain data. The reason why the illegal connection attack is possible is that the pairing method between smart wearable devices and smartphones is not by authentication. According to the study, special unauthorized Apps can be installed on Android 4.3, which can pair with smart devices from specific manufacturers. The user needs to confirm the pairing before establishing a connection. However, the victim cannot know whether it is associated with his/her device or someone else’s device.Unauthorized control: Studies have shown that an attack A can send a specific command to the Lenovo watch x to set multiple alarms or forge a call alert because of lacking user rights’ control [42]. Under normal circumstances, the user can only send information and instructions to the bound smart watch by himself/herself. However, some users also can operate the unbound watches, because the cloud server does not judge the identity and the instructions.Private data leakage: We assume that the cloud and the data requester are honest, but curious; this means they would obey the workflow, but want to obtain other entities’ private information. Meanwhile, it is surprising that most transmission data channels between smart watches and the cloud server are public and unencrypted [43]. An attacker can obtain user information by sniffing. Moreover, even if user data traffic is encrypted, the cloud server can attempt to decrypt with a powerful computational capability. Besides, other users attempt to use their own attributes to obtain the private key for decrypting information.

A summary of the attack model is shown in Figure 3.

### 4.2. System Model

For resisting the above attacks, we propose a system consisting of three parts: a smart watch, a management App, and the cloud. The user involved in the design also includes the primary user and the requesters with different attributes.

A smart watch is a smart terminal that collects wearer’s health information such as heart rate, number of steps, etc. In addition, it can also implement the positioning function.

The management App is an application for managing smart wearable devices. Not only can it show the user’s health information, but also, it can know the location of the wearer by the location of the smart wearable device. At the same time, it can implement information sharing and access control according to different user groups.

The cloud is responsible for the complex calculation.

The wearer of the smart watch usually is the primary user. He/she can selectively share permissions with other users such as his/her friends or other relatives through the management App authorization. A summary of the system model is shown in Figure 4.

### 4.3. Working Process

This subsection introduces the workflow of our scheme.


**Phase 1: Initialization, in which the App establishes a connection with the smart watch and negotiates a key with the cloud.**


1:A smart watch connects with the management App through scanning the quick response code (QR code) in the smart watch. The App obtains the MAC value and UUID of the smart watch to form a unique smart watch identity through the communication protocol. The App uses the hash function and saves the unique identity in the list. The App generates a key pair and sends the public key to the smart watch.2:It would generate new keys unless the smart watch disconnects. If repairing, the APP must authenticate the MAC and UUID of the watch. If the unique identity matches, proceed to Step 1.3. Otherwise, the device is considered unsafe and disconnects.


**Phase 2: The smart watch collects and processes the data.**


1:The smart watch collects location information and personal health information of the wearer, including steps walked and heart rate, and encrypts the data by the public key of the App.2:The smart watch sends the encrypted data to the App.


**Phase 3: The App and cloud process the data.**


1:After receiving the data, the primary user can change the encrypted data by $pk_{APP}$ to the encrypted data by PK and send it to the cloud.2:The cloud adopts the homomorphic computation and re-encryption for obtaining the final computing result ${\left[m\right]}_{re-enc}$. The re-encryption step can transfer the ciphertext that the primary user can decrypt.3:The cloud sends the computing result ${\left[m\right]}_{re-enc}$ to the App.4:The App can decrypt the computing result ${\left[m\right]}_{re-enc}$ by his/her private key.


**Phase 4: Access control of the App.**


1:The secondary user applies the data access authority.2:In order to ensure that the shared data can only be accessed by the authorized requester by the primary user, the primary user encrypts the decryption key SK through the ciphertext policy based on the access control tree. Then, the App sends the encrypted key CT to the requester.3:The cloud sends the ciphertext, which is homeomorphically computed and re-encrypted, to the requester.4:The requesters first decrypts the ciphertext CT and gets the key SK only if their own attribute set satisfies the access control structure. Then, the requesters decrypt the final computing result ${\left[m\right]}_{re-enc}$ when they obtain SK.

The entirety of the data is ciphertext in transmission, so they can effectively prevent information leaking and from being obtained by an unrelated person. Figure 5 shows the workflow of our scheme.

### 4.4. Concrete Construction

To achieve the security requirements, the scheme consists of four parts:

Part 1: System initialization

***System setup:** All entities except the smart watch call the algorithm KeyGen. The APP and the users call the algorithm CP-ABE-setup to complete the initialization of the homo-re-encryption and CP-ABE.***KeyGen:** Let *k* be a security parameter and *p*, *q* be two large primes. Due to the property of safe primes, we can choose two primes $p^{\prime }$ and $q^{\prime }$, which satisfy $p=2p^{\prime }+1,q=2q^{\prime }+1$. We compute n=p×q and choose a generator *g* with order $\lambda =2p^{\prime }q^{\prime }$, which can be chosen by selecting a random number $z\in Z_{n^{2}}^{*}$ and computing $g=-z^{2n}$. The value λ can be used to decrypt the encrypted data, but we decided to conceal it from the whole system. In FPAS, we can use key pair $\left(sk,g^{sk}\right)$ to encrypt and decrypt data. The App and the CCS generate their key pairs: $\left(sk_{App}=a,pk_{App}=g^{a}\right)$ and $\left(sk_{CCS}=b,pk_{CCS}=g^{b}\right)$ and then negotiate their Diffie–Hellman key $PK=pk_{App}^{sk_{CCS}}=pk_{CCS}^{sk_{App}}=g^{a*b}$. To support encrypted data processing, PK is public in the whole system. The public system parameters include {g,n,PK}.***CP-ABE-Setup:** The setup algorithm chooses a bilinear group $G_{0}$ of prime order *p* with generator *g*. Then, it picks two random exponents $\alpha ,\beta \in Z_{p}$. The public key is $PK=G_{0},g,h=g^{\beta }$, and the master key MK is $\left(\beta ,g^{\alpha }\right)$.

Part 2: Translation from the smart watch

***System matching:** After the smart watch connects with the App, the smart watch sends Hash(MAC∥UUID) to the App. the App saves it into a list.***Watch-KeyGen:** The APP calls the algorithm that lets α, β be two large primes and calculates γ=α×β and φ(γ)=(α−1)(β−1). Let eandφ(γ) be co-prime. Then, choose *d* such that ed≡1. The public key $pk_{APP}$ is (γ,e), and the private key $sk_{APP}=\left(\gamma ,d\right)$. The App sends $pk_{APP}$ to the smart watch.***Watch-Enc:** The smart watch uses the public key $pk_{APP}$ to encrypt *m* and send it to the App.
(1)$$\begin{array}{cc} {\left[m\right]}_{pk_{APP}} & =\{T,T^{\prime }\}\\ & =\{\left(1+m\times n\right)\times pk_{APP}^{r},g^{r}\} \end{array}$$

Part 3: Data processing of the CCS

***Data uploading:** The App can change the encrypted data ${\left[m\right]}_{pk_{APP}}$ to ${\left[m\right]}_{PK}$ by choosing a random r∈[1/n] and sending it to the cloud:
(2)$$\begin{array}{cc} {\left[m\right]}_{PK} & =\{T,{\left(T^{\prime }\right)}^{sk_{APP}}\}\\ & =\{\left(1+m_{i}\times n\right)\times PK^{r},g^{r\times a}\} \end{array}$$***Data processing:** After receiving the encrypted data from the App, the CSSdoes some homomorphic comparison operations to get the ciphertext results. If the encrypted data exceed an encrypted threshold [t], this means the CCS needs to send a warming and the re-encrypted result ${\left[m\right]}_{pk_{re-enc}}$ to the primary user and other authorized users. The data are encrypted by the App by choosing a random $r_{2}\in \left[1/n\right]$ as follows:
(3)$$\begin{array}{cc} \left[t\right] & ={\left[t\right]}_{PK}=\left\{T_{2},T_{2}^{\prime }\right\}\\ & =\{\left(1+t\times n\right)\times PK^{r_{2}},g^{r_{2}}\} \end{array}$$
The homomorphic cloud computes the result:
(4)$$\begin{array}{cc} {\left[m\right]}^{*} & =\{T,T^{\prime }\}\\ & =\{T\times T_{2}^{n-1},T^{\prime }\times {\left(T_{2}^{\prime }\right)}^{n-1}\}\\ & =\{\left(1+m\times n\right)\times PK^{r}\times \left(1-t\times n\right)PK^{r_{2}\left(n-1\right)},g^{r}\times g^{r_{2}\left(n-1\right)}\}\\ & =[(m)-t] \end{array}$$
Then, [m] is changed into another ciphertext that can be decrypted by the primary user and other data requesters who are authorized through the algorithm:
(5)$$\begin{array}{cc} {\left[m\right]}_{re-enc} & =\left\{T^{\left(1\right)},T^{\prime \left(1\right)}\right\}=\left\{T,T^{\prime {sk}_{CCS}}\right\}\\ & =\{\left(1+m_{1}\times n\right)\times PK^{r_{1}}\times \left(1-t\times n\right)\times PK^{r_{2}\left(n-1\right)},g^{b(r_{1}+r_{2}\left(n-1\right))}\} \end{array}$$
Finally, send the ${\left[m\right]}_{re-enc}$ to the parents and the authorized users.

Part 4: Access control of the App

***Accessing authorization:** The data owner of the App just wants to share his/her personal information with someone who satisfies his/her access condition. He/she will encryption his/her private key $sk_{pa}$ with the encryption algorithm CP-ABE.
(6)$$\begin{array}{c} Encrypt(PK^{\prime },a,A)=CT\\ =(A,C=ae{\left(g,g\right)}^{\alpha s},C=h^{s},\\ \forall y\in Y:C_{y}=g^{q_{y}\left(0\right)},C_{y}^{\prime }=H{\left(att\left(y\right)\right)}^{q_{y}\left(0\right)}) \end{array}$$
The CCS sends the CT to the primary user.***Data acquisition:** When the primary user receives the outsourced computing result, he/she can decryption directly ${\left[m\right]}_{re-enc}$ by his/her private key $sk_{FG}$.
(7)$$m=L\left(T^{\left(1\right)/{\left(T^{\prime \left(1\right)}\right)}^{sk_{FG}}}\right),\mathrm{where}\;L\left(X\right)=\left(X-1\right)/n$$
The dependent users desire to obtain the result by decrypting CT to get the private key of the dependent users. Firstly, dependent users recall the key generation algorithm.
(8)$$\begin{array}{c} KeyGen(MK,S)=SK=(D=g^{(\alpha +r)/\beta },\forall j\in S:D_{j}=g^{r}\cdot H{\left(j\right)}^{r_{j}},D_{j}^{{}^{\prime }}=g^{r_{j}}) \end{array}$$
Secondly, the secondary users recall the decrypt algorithm to decrypt CT, and they first compute:
(9)$$\begin{array}{cc} B & =Decryptnode(CT,SK,r)\\ & =e{\left(g,g\right)}^{rq_{R}\left(0\right)=e{\left(g,g\right)}^{rs}}. \end{array}$$
If the secondary user’s attributes satisfy the access structure, the algorithm now decrypts by computing:
(10)C/(e(C,D)/B)=a
Then, the dependent users call the decryption algorithm:
(11)$$\begin{array}{c} m=L(T^{\left(1\right)}/{\left(T^{\prime \left(1\right)}\right)}^{a}) \end{array}$$

## 5. Security Analysis

Based on the existing attack model defined in Section 4.1, we conducted a theoretical analysis of the security of the scheme, trying to prove that our scheme can resist existing attacks. We divided it into three parts and analyzed the security of FPAS.

First, if the attacker A launches the illegal application connection attack, we can prove our scheme can resist the attack. We set the PIN value during the device pairing phase of the protocol. The PIN code is displayed on the application’s screen when the user’s application wants to connect to the smart watch’s App. If A attempts to connect to the device using a third party application, A cannot guess the PIN on the smart watch screen. The PIN was a random combination of four digits and letters, and A guesses the probability of success $1/4^{36}$. The possibility of successful attacks can be ignored, so we can say that our solution can effectively prevent attackers from silently connecting malicious devices. Moreover, if the smart watch uses the Bluetooth protocol in connection, the legitimate application will record and save the MAC and UUID in the smart watch as a unique identity of the watch the first time. The data encryption algorithm added by the protocol is also based on MAC and UUID identity encryption. Therefore, it checks whether the MAC and UUID match with the recording in the application. Since the MAC and UUID are unique in the Bluetooth protocol, the device identification number is confirmed. A can use a malicious device, but it is hard to falsify MAC and UUID and establish a connection. It can avoid the smart watch as a secret channel to reveal important data on the mobile phone.

Secondly, if A gets some ciphertext and tries to decrypt it in an unauthorized situation, we can analyze that A cannot obtain some privacy information. We have established a strict user rights’ control mechanism in the program. Ciphertext-based attribute-based encryption is used during the data sharing phase. The decrypted access control tree is determined by the encryption side. Without satisfying the authorized user access structure, other users cannot use their own attributes to obtain the decrypted private key. Therefore, other curious users only can decrypt information under the control of the primary user. Moreover, in this case, a smart watch is bound to multiple Apps, for example a children’s watch can be bound to different persons such as parents and teachers. We used re-encryption to convert the decryption key according to the attributes of different visitors, which has reached the purpose of fine-grained access control. Fang et al. [44] have proven that the proxy re-encryption algorithm can resist against the chosen ciphertext attacks in ciphertext sharing. Therefore, we believe that even if the cloud is not completely trusted, our solution still does not lead to data leakage, and the private data are not intercepted by unauthorized users.

Finally, if the attacker A is the cloud, that means the cloud is honest, but curious [45]. In our solution, data are encrypted in the transaction. If A intercepts the data and cannot obtain the decrypted private key without being authenticated, the plaintext information cannot be recovered. From smartphones to cloud providers, we used re-encryption with homomorphic features. Similarly, an attacker cannot guess the private key to recover the plaintext. The security of our scheme is based on the security of the encryption algorithm [46]. Therefore, the scheme can resist the untrusted cloud server accessing private data.

## 6. Performance Analysis

In this section, we analyze the computational complexity of our scheme.

**Computational complexity.** We analyzed the computational complexity of CP-ABE. First, we based this on the assumption that |u| universal attributes exist, |a| attributes are in the access tree *A*, and that at most *s* attributes satisfy the access tree *A* for decrypting [30]. For the setup algorithm CP-ABE-Setup, it needs to do one exponentiation and one bilinear pairing. The encryption algorithm CP-ABE-Encrypt needs to encrypt the message by each attribute in *A* with two exponentiations and encrypt the message with one exponentiation and one bilinear pairing. Therefore, it needs to do 2|a|+1 exponentiations and one bilinear pairing in total. The key generation algorithm CP-ABE-KeyGen also does 2|a|+1 exponentiations and one bilinear pairing. CP-ABE-Decrypt operates at most *s* bilinear pairings and *s* exponentiations to decrypt the ciphertext. Though the computational cost of CP-ABE is higher than KP-ABE [30], it has safer performance, and the primary user can control the access authority. Considering the improvement on the security level, this amount of overhead is reasonable.

As we know, modular exponentiation spends more time than the addition and multiplication operation, so we can overlook the number of addition and multiplication operations in our next analysis. If the computation complexity is not related to the number of computed data, the computational complexities can be set to O(1). We have a hypothesis that the quantity of data is *N*in our scheme. In the data uploading process, it needs to operate 2N exponentiations and results in computational complexity O(N). The process of data processing needs eight exponentiations, so the computational complexity is O(N).

**Overhead in FPAS.** Next, we do some experiments to detect the overhead of our scheme, and the nodes include the App, smart watch, and cloud. We adopt the i5-7400, 3.00 GHz, 8.00 GB, and Windows-64-bit laptop to simulate the computing environment. For the smart watch, we use the Qualcomm Snapdragon Wear 2100 chip with 1.2 GHz for simulation, which is more common in a smart watch. For the smartphone, we use the Android 7.0 64-bit operating system with 1.5 G of memory. Through the analysis of the program, we find that the operation of the entire protocol could be divided into three parts: encryption data in the smart watch, processing data of the CCS, and access control of the data results by the App. Assuming the number of requester’s attributes |Q| is two and the length of the data *n* is 1024 bit, the overhead of our scheme is shown in Table 2. From Table 2, we infer that the overhead of the smart watch accounted for about 22% of the total system overhead. Even though the computing power is weaker relative to other devices, it does not spend more time in encryption. Moreover, we allocate the computing to the cloud from the smartphone, which reduce the computing overhead of the mobile phone and improve the system efficiency under the premise of security.

The entire protocol overhead is mainly generated by these three parts mentioned above. The overhead of transmitting data is mainly generated by the data encryption watch-enc. The data generally is health data or user location information monitored by the smart watch. This time is closely related to the data length *n*. Therefore, we could perform a time comparison by setting the data length to (512, 768, 1024, 1280, 1536, 1792, 2048) bits, respectively. The results are shown in Figure 6. Although the overhead is increasing with the length of the data, the data such as steps or heart rate are produced intermittently and are not too large. Hence, the overhead is still affordable. We can ignore the time when the encrypted data were uploaded to the cloud computing server. We pay more attention to the time overhead of the cloud computing server for data processing, including the encrypted data homomorphic operation and the re-encrypted computing part. This time of re-encrypted computing was also related to the length of the data. The overhead results are shown in Figure 6. Similarly, the overhead of FPAS is acceptable for mobile phones and negligible for the cloud. As for the time spent by the App controlling the access rights, this was mainly generated by the CP-ABE. Depending on the granularity of the permissions, different numbers of attributes |Q| could be used to describe the identity of a user. Different numbers of attributes bring different time overhead. We set the number of attributes |Q| as (1, 2, 3, 4, 5) for testing the time required in our scheme. We divide the whole process into encryption, different decryption keys’ generation, and decryption according to different attributes. The results are shown in Figure 7.

From the above experiments, we infer that the overhead of our system FPAS mainly come from encrypting data by the smart watch and uploading data and accessing control by the App. Since the actual computing power of the cloud server is stronger than the computing power of the PC simulation, we can ignore the time overhead in the actual cloud computing server. Moreover, the most overhead of the homomorphic computing and re-encryption processes in the entire solution is carried out in the cloud, so we pay more attention to the computing power of smart watches and smartphones. Taking the Android watch as an example, the CPU frequency of the Qualcomm Snapdragon Wear 2100 chip is 1.2 GHz. From the experimental results, we infer that the calculation speed would be gradually increased with the increasing of the CPU frequency. Therefore, we believe that the overhead of our solution depended on the industry level of existing chips. At the same time, with the improvement of the smartphone’s computing performance, the overhead is decreasing. Through the experimental results, we can understand that the overhead of our scheme is mainly concentrated in data processing and access control. However, the implementation of processing data and access control mainly relies on mobile Apps and cloud servers. Currently, some chips used in IoT systems have low computing power because the requirements of computing are simple. Hence, in the IoT system, many devices have low computational power and have difficulty achieving high security. The FPAS can balance security and computing power, so the FPAS is applicable to other IoT systems. Furthermore, our scheme does not depend on the hardware environment. Hence, it is easy to apply to other smart devices.

**Comparison.** We compare our scheme with [13,25,30,31] in term of security and functionality in Table 3. The existing smart watch security protocols rely on a trusted cloud server. The solution includes various strategies for solving personal sensitive data leakage of the cloud platform. For example, it uses attribute encryption to realize controllable access of data, homomorphic encryption for confidential calculation on encrypted private data, and the proxy re-encryption policy for encrypted data sharing. However, the cloud server and the user are always in different trust domains. Therefore, to reduce trust on a third cloud server, a secure fine-grained information sharing scheme is proposed in this situation. From Table 3, we find that FPAS more comprehensively addresses the security issues of privacy protection and data sharing in the context of smart watches. It is worth mentioning that we also are concerned with the problem of illegal application connections that are ignored by other solutions. We use the unique identification code generated by the smart watch and the application to confirm the uniqueness of both devices, avoiding the illegal connection of malicious devices or malicious programs. Moreover, we still can safely perform operations as FPAS reserves part of computing in the App and the private keys are not sent to the untrusted cloud environment. FPAS does not face the problem of excessive permissions for the cloud.

## 7. Conclusions

In this paper, we proposed FPAS, a fine-grained access control protocol for information content and access rights when the cloud is not trusted. The solution is compatible with devices that have different computing power. Moreover, we analyzed that our protocol is application agnostic and can be applied in other IoT systems since the underlying protocol does not depend on a trusted hardware environment. FPAS improves efficiency and prevents unauthorized access. Through theoretical analysis and experiments, FPAS had reliable performance and effective data processing capabilities. Hence, FPAS is suitable for smart wearable device and also has a good expansibility.

## Figures and Tables

**Figure 1 sensors-19-02109-f001:**
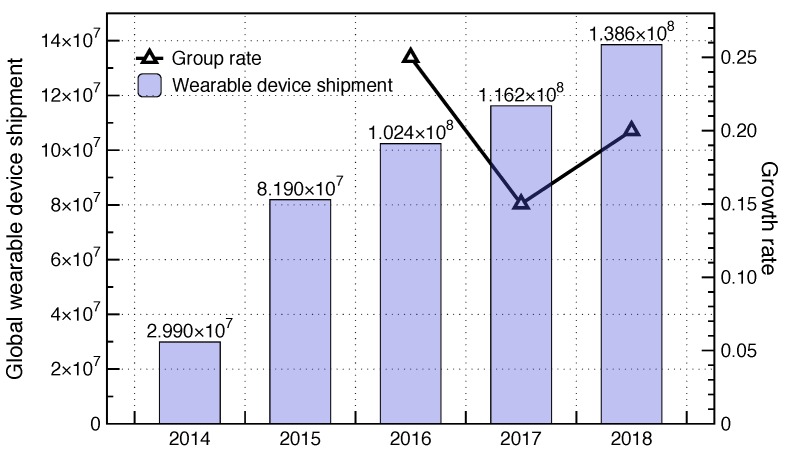
The sales of smart watches.

**Figure 2 sensors-19-02109-f002:**
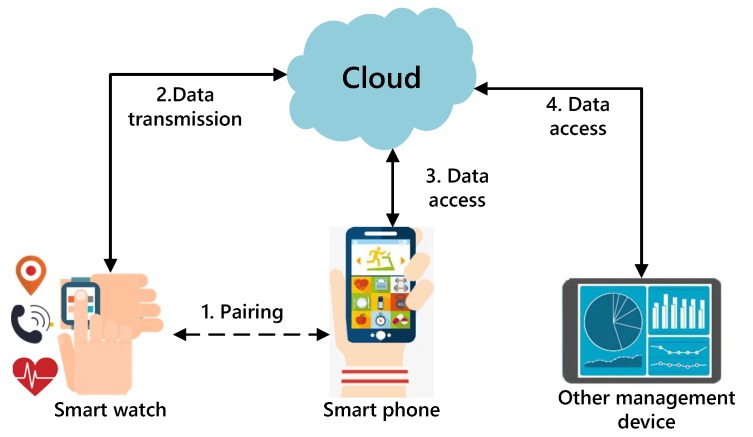
The information interaction process of the smart wearable device.

**Figure 3 sensors-19-02109-f003:**
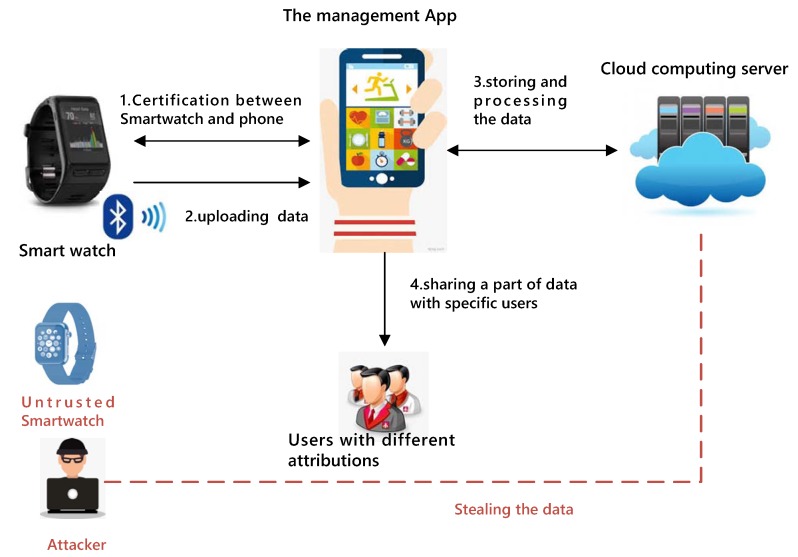
Attack model.

**Figure 4 sensors-19-02109-f004:**
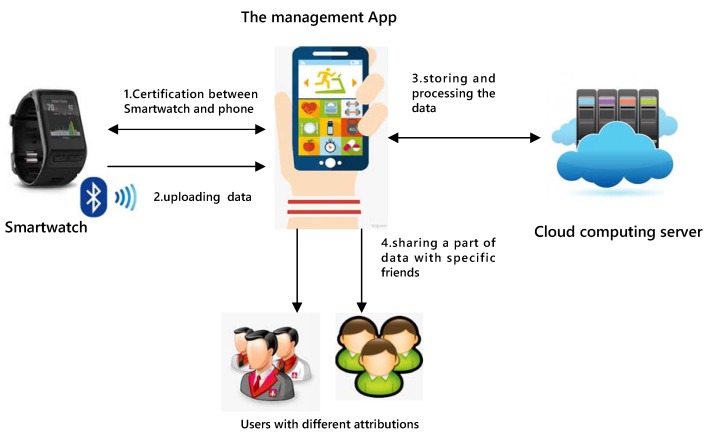
System model.

**Figure 5 sensors-19-02109-f005:**
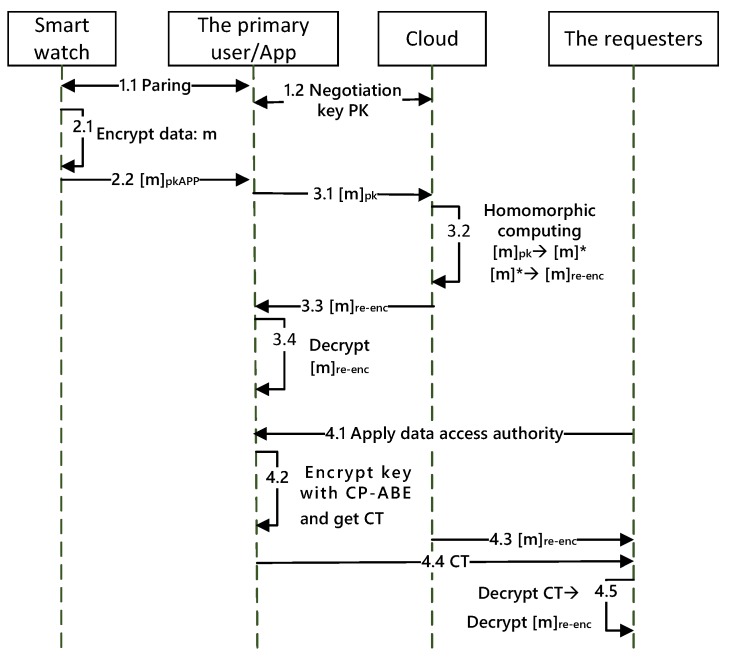
The data flow of FPAS.

**Figure 6 sensors-19-02109-f006:**
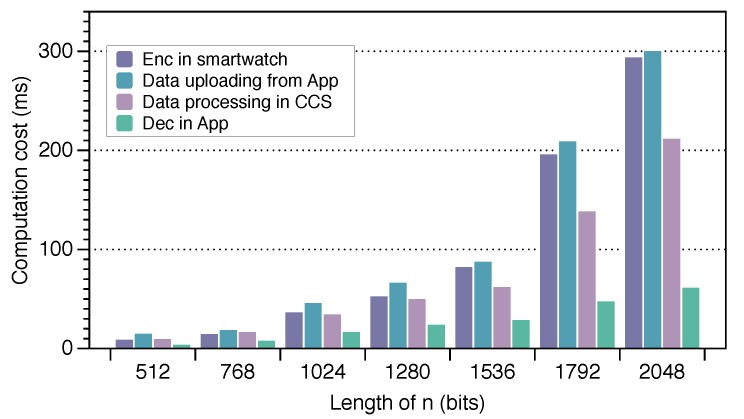
The time spent on data transmission and data processing.

**Figure 7 sensors-19-02109-f007:**
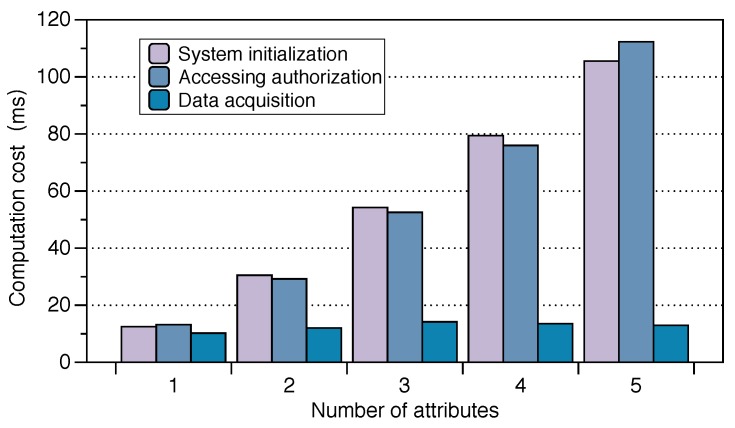
The time spent on accessing control.

**Table 1 sensors-19-02109-t001:** Symbols and their meanings.

Symbols	Explanations
*k, p, q, n*	The system parameters;
(sk,pk)	The key pair of data
$pk_{APP}$	The public key of the App
PK	
$=pk_{APP}^{sk_{CCS}}$	The public key of the primary user and CCS
$=pk_{CCS}^{sk_{APP}}$	
*m*	The data collected by the smart watch
${\left[m\right]}_{pk_{APP}}$	The ciphertext of *m* under $pk_{APP}$
${\left[m\right]}_{PK}$	The ciphertext of *m* under PK
[m]*	The ciphertext of *m* under homomorphic computing
${\left[m\right]}_{re-enc}$	The re-encrypted ciphertext of *m*
*t*	The threshold of the system
SK	The decryption key
CT	The encrypted key by CP-ABE
*N*	The quantity of data
*n*	The length of data
|Q|	The number of attributes

**Table 2 sensors-19-02109-t002:** FPAS overhead (*n* = 1024 bit, |Q|=2).

	Qualcomm Snapdragon Wear 2100 Chip	Smart Phone with Android 7.0	Cloud
Encrypt the data	48.12 ms	56.78 ms	5.32 ms
Homomorphic computing	−	−	31.40 ms
Decrypt the data	−	24.56 ms	−
Re-encrypt the data	−	−	13.42 ms
Key generation	−	30.52 ms	−
Total time	48.12 ms	111.86 ms	50.14 ms

**Table 3 sensors-19-02109-t003:** Comparison with different schemes.

	Liu et al.	Ding et al.	Khedr et al.	Zhang et al.	Our Work
	[31]	[30]	[25]	[13]	
prevent illegal application connection	−	−	−	−	*√*
resist unauthorized attack	×	*√*	×	×	*√*
resist sniffing data attack	×	×	×	*√*	*√*
security ciphertext computing	×	*√*	*√*	×	*√*
fine-grained user-divided	*√*	*√*	×	*√*	*√*
secure data sharing	*√*	*√*	*√*	*√*	*√*

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
