# Peer review of "A Fine-Grained User-Divided Privacy-Preserving Access Control Protocol in Smart Watch"

_sensors, 2019, doi:10.3390/s19092109_

Round 1
Reviewer 1 Report
This paper presents a fine-grained privacy-preserving access control architecture for smart watches to mitigate the balance between the privacy-preserving and data access control. The work is well motivated and the manuscript makes technical sense. The paper is well organized and materials are well presented to support its idea. However, I did not see much scientific contribution to the community and the paper looks more like an engineering report. Does it have potential opportunities to be used in other IoT problems.
Author Response
Point 1: This paper presents a fine-grained privacy-preserving access control architecture for smart watches to mitigate the balance between the privacy-preserving and data access control. The work is well motivated and the manuscript makes technical sense. The paper is well organized and materials are well presented to support its idea. However, I did not see much scientific contribution to the community and the paper looks more like an engineering report. Does it have potential opportunities to be used in other IoT problems.
Response 1: Thank you for the comments. In this revised version, we described the scientific contributions in detail. In the existing smart watch security protocols, they rely on a trusted cloud server. However, the cloud server and the user are always in different trust domains. Therefore, to reduce trust on a third cloud server, proposed a secure fine-grained information sharing scheme in this situation. This fine-grained access control not only refers to restrictions on persons of private data accessing, but also to partial sharing of personal information. Moreover, we also consider the differences in computing power of smart watch and proposed a privacy protection scheme for fine-grained access control of information content and access rights when the cloud is not trusted. The solution is compatible with devices that have different computing power. Moreover, we analyzed that our protocol is application agnostic and can be applied in other IoT systems since the underlying protocols do not depend on trusted hardware environment. We addressed this comment in Lines 79-92 page 3.

Reviewer 2 Report
The authors propose a fine-grained privacy-preserving access control architecture for smart watches.
The paper is well organized. Figures and tables are appropriate and help to understand the concepts presented.
Suggestions for Improvement:
1- Extensible English corrections;
2- Line 32 does not make any sense: "It needs..."
3- Line 34: "For the...third user"; Which mechanism? The paragraph (34 to 42) needs to be rewritten.
4- Line 54: Bluetooth should carry out this task!
5- Line 55: A traditional password scheme would prevent data leakage!
6- Line 56: A traditional authentication scheme would prevent this!
7- Line 57: "weak"? OR low processing capacity (what is low processing capacity nowadays?!)
Author Response
Response to Reviewer 2 Comments
Point 1:The authors propose a fine-grained privacy-preserving access control architecture for smart watches. The paper is well organized. Figures and tables are appropriate and help to understand the concepts presented.
Response 1: Thank you for this comment. We would like to thank the reviewers for their recognition of our work.
Point 2: 1- Extensible English corrections
Response 2: Thank you for this comment. In this revised version, we checked the paper from
scratch and proofed our English.
Point 3: 2- Line 32 does not make any sense: ”It needs...”
Response 3: Thank you for your suggestions. We have removed it in the revised version.
Point 4: 3- Line 34: ”For the...third user”; Which mechanism? The paragraph (34 to 42) needs to be rewritten.
Response 4: Thank you for this comment. We have revised our statement to clarify this issue. The mechanism refers to the working mechanism of the smart wearable device, similar to the information interaction process mentioned in Figure 2. This paragraph mainly describes the weak points in the information interaction mechanism of smart watch, which can be exploited by an attacker. The attack strategies include malicious connection, illegal data processing and unauthorized accessing. We have rewritten the paragraph in the revised version of this paper in Lines 34-43 page 2.
Point 5: 4- Line 54: Bluetooth should carry out this task!
Response 5: Thank you for this comment. As you said, Bluetooth should carry out this task. However, some researchers have found that a critical cryptograohic vulnerability has been found affecting the Bluetooth that could allow an unauthenticated [1]. In addition to the potential vulnerabilities in the Bluetooth protocol, we also pay attention to the security risks of the application. Some smart watch manufacturers have developed applications for connecting smart watches. Due to the lack of uniform security standards or security awareness, attackers may bypass the authentication mechanism and cause a malicious connection to devices. Hence, we have better adding extra protocol to ensure the security under existing design flaw of Bluetooth. We have added this statement to clarify the contribution of our paper in Line 55 page 3.
Point 6: 5- Line 55: A traditional password scheme would prevent data leakage!
Response 6: Thank you for pointing this out. The traditional cryptographic protocols can partially solve the problem of information leakage, in our scheme, however, the cloud is not always credible, it may be honest but curious and cloud needs to calculate the data. Obviously, the cloud should not have the right to decrypt the ciphertext, otherwise, it causes more serious information leakage. Therefore, we need to propose a secure fine-grained information sharing scheme in this situation. This fine-grained access control not only refers to restrictions on a person of private data accessing, but also to partial sharing of personal information. So a security scheme that can conduct confidential calculations should be proposed even though the cloud is not trusted. We have modified the statement in the revised version paper in Line 57 Page 3.
Point 7: 6- Line 56: A traditional authentication scheme would prevent this!
Response 7: Thank you for this comment. Traditional identity authentication protocols prevent unauthorized access. However, traditional identity authentication fails to solve the problem of reducing interaction in information sharing. Specifically, we prefer those information owners to participate in the allocation of access control permissions, but in the information sharing phase only relying on the untrusted cloud without information leakage. If the data owner completely sets access control permissions based on the visitors one by one, it would overly consume the device’s computing resources and cause unnecessary computational overhead. Therefore, the problem we have to solve is to reduce the ciphertext sharing steps as much as possible under the premise of fine-grained access control. It takes into account the privacy sharing of personal information and also reduces the computational overhead of low-computing devices. We have modified the expression in the revised version in Line 58 Page 3.
Point 8: 7- Line 57: ”weak”? OR low processing capacity (what is low processing capacity nowadays?
Response 8: Thank you for this comment. In fact, the”weak computing device” should be changed to ”low processing device”. Nowadays, some chips used in IoT system have low computing power because the requirements of computing are simple. Hence, in the IoT system, many devices have the low computational power and are hard to achieve high security. Therefore, we hope the proposed protocol can balance security and computing power. It will be better if the FPAS is applicable to other IoT systems. In this revised version, we provided an explanation. We have modified the expression in the revised version in Line 60 Page 3. Thanks again.
References
[1] New bluetooth hack affects millions of devices from major vendors. https://thehackernews. com/2018/07/bluetooth-hack-vulnerability.html.

Round 2
Reviewer 2 Report
The authors addressed all modifications pointed out by the reviewer.